# Preliminary Design and On-Site Testing Methodology of Roof-Cutting for Entry Retaining in Underground Coal Mine

**DOI:** 10.3390/s23146391

**Published:** 2023-07-14

**Authors:** Ying Chen, Zikai Zhang, Shiji Bao, Hongtao Yang, Mingzhe Shi, Chen Cao

**Affiliations:** College of Mining, Liaoning Technical University, Fuxin 123000, China; zhangzikai7626@163.com (Z.Z.); sj18342865602sj@163.com (S.B.); yanghongtao2023@163.com (H.Y.); shimingzhe2023@163.com (M.S.); caochen@lntu.edu.cn (C.C.)

**Keywords:** entry retaining, roof-cutting blasting, field study, preliminary design, dynamic monitoring

## Abstract

Entry retaining via roof cutting is a new longwall mining method that has emerged in recent years, and is characterized by high resource utilization and environmental friendliness. Due to the complexity of this method, a field study is commonly employed for process optimization. Roof blasting is a key operation for retaining the entry, and the current practice involves dynamically adjusting blasting parameters through on-site testing and postblasting monitoring. However, the existing literature lacks detailed descriptions of blasting operations, making it difficult for field engineers to replicate the results. In this study, based on a roof cutting project for entry retaining, a preliminary design of blasting parameters is made based on theories and on-site geological conditions. The on-site test methods and equipment for roof-cutting blasting are described in detail, and the fractural patterns under different blasting parameters are analyzed. After the retreat of the working face, the state of roof caving in the goaf is analyzed based on monitoring data, and the effectiveness of top cutting is evaluated through reverse analysis, leading to dynamic adjustments of the blasting parameters. This research provides a reproducible construction method for roof-cutting operations and establishes the relationship between blasting parameters and post-mining monitoring data. It contributes to the development of fundamental theories and systematic technical systems for entry retaining via roof cutting, offering high-quality case studies for similar geological engineering projects.

## 1. Introduction

The longwall mining method using double entries is a traditional underground coal mining method and is currently the most common one. However, this method leaves a coal pillar for each panel, resulting in a resource utilization rate of about 50% [1,2]. The residual coal pillars cause stress to be concentrated underground, posing safety risks in multi-seam mining. Moreover, the residual coal pillars lead to uneven subsidence of the overlying strata, resulting in permanent environmental damage, such as groundwater loss and ground fractures, with little possibility of ecological restoration [3,4,5,6].

To achieve sustainable development of underground coal resources, a longwall mining method of entry retaining via roof cutting has been increasingly applied in engineering practice in recent years [4,5,6,7,8,9,10]. This method utilizes the goaf gangue as one entry rib, eliminating the natural coal pillar or an artificial pillar and greatly improving resource utilization and environmental friendliness. However, this method involves complex processes and technical difficulties. The entry-retaining technique shortens the hanging roof length in the goaf through pre-split blasting on the roof ahead of the working face. After the mining of the working face, the goaf gangue collapses along the pre-split plane to form a gangue support. The entry is preserved, serving as the next working face.

The key technology of entry retaining is the bi-directional blasting of the roof. The blasting involves detonating explosives in a selected direction using an energy accumulation device set, which causes tensile stress in the surrounding rock in the designated direction and compressive stress in the non-designated direction, thus inducing a fracture in the rock in a selected direction. The effect of roof cutting plays a decisive role in the stability of the retained entry [10,11,12]. The technical challenge lies in the requirement to create continuous fracture surfaces in the goaf side roof, facilitating thorough roof caving after mining, while avoiding damage to the entry roof and preserving its integrity as much as possible. Small variations in the roof’s lithology, strength, and thickness can affect the roof-caving results [12,13,14,15]. Therefore, accurate detection of the roof’s lithology, strength, and thickness is necessary before blasting, and should be followed by on-site testing to determine the blasting parameters under different conditions. After mining, the collapsed state of the roof is analyzed based on monitoring data, and the effectiveness of roof cutting is evaluated through reverse analysis, allowing for dynamic adjustments and optimization of the blasting parameters.

Due to the complexity of the entry-retaining method, existing research has mainly relied on case-based on-site studies [4,5,6,7,8,9,16,17,18,19,20,21,22,23,24,25]. However, these studies often lack sufficient details regarding the blasting design and on-site testing techniques. The descriptions in the existing literature are often brief, making it difficult for on-site engineers to replicate the results. Furthermore, the analysis of blasting effectiveness heavily relies on the field experience of construction personnel, and there are deficiencies in post-blasting data monitoring and analysis. Establishing a systematic design principle and engineering practice methodology for roof blasting is of great significance for promoting the application of the entry-retaining method and achieving efficient resource utilization. This paper takes the one working face in the Donghuantuo Coal Mine as the engineering background, conducts a detailed analysis of roof-blasting operations, utilizes theoretical analysis methods to obtain preliminary designs, analyzes the blasting effects through on-site experiments, and dynamically adjusts and optimizes the blasting parameters based on post-blasting data monitoring and analysis. This research provides high-quality case studies for establishing fundamental theories and systematic technical systems for roof-cutting blasting, serving as a reference for similar geological engineering projects.

## 2. Materials and Methods

### 2.1. Engineering Background

The Donghuantuo Coal Mine is located northwest of Tangshan, Hebei, China. It has a total of 9 coal seams, with coal seams 8, 9, 11, and 12-1 being the main ones used for mining. The 20221 working face is in the southern wing of the central mining area at about 800 m underground and develops along the 12-2 coal seam. The northern end of the working face is connected to the coal 12-2 belt tunnel of the central mining area, while the southern end extends 320 m south of the 14# exploration line. The western side is the unmined 20223 working face, and the eastern side consists of the returning airway and the auxiliary airway. Above the panel are goafs of the 8#, 9#, 11#, and 12-1# coal seams and the 2312 working face. The main gate for the 20221 panel is 1850.4 m long, and the working face is 200 m long. The layout and retained entry positions of the 20221 working face are shown in Figure 1.

The 20221 working face mines the 12-2# coal seam, which is stable sedimentation. The thickness of the coal seam shows a trend of thickening to the north and thinning to the south in this area, and the structure is complex. There is a layer of composite coal seam that has developed above and below the main coal seam, with a thickness of about 0.2 m. The coal seam has a metallic luster and is bright and gaseous in nature, with an average thickness of 1.88 m. For details, refer to Table 1 and Table 2.

### 2.2. Roof Detection

A 10 m deep borehole was drilled in the roof of the 20,221 haulage roadway every 200 m. A ZKXG100 mine-drilling imaging trajectory detection device which is produced by Wuhan Tensense Geotech Prospecting Technology Co., Ltd., Wuhan, China, was used for borehole observation, as shown in Figure 2.

Figure 3 shows the borehole observation of the weak layer and the basic roof condition near the 1500 m advancing distance. It is worth noting that there is a weak layer 2–3 m above the coal seam. Based on the analysis of the geological column, this rock layer is identified as gray-black siltstone. In some borehole observations, the color of the rock layer changes from dark gray to light gray, indicating the presence of fine sandstone.

Assuming that there are no undulations in the roof of the 20221 main gate in the strike direction, a profile of the overlying strata of the 20221 haulage roadway was plotted as shown in Figure 4.

### 2.3. Roof-Cutting Design

#### 2.3.1. Cutting Height

The roof-cutting height mainly considers the filling of the gangue after roof caving and the need to provide support for the roof of the goaf. Based on the roof conditions shown in Figure 5, the roof-cutting height is calculated as follows [1,4]:(1)$$H_{Q}=\frac{M}{\left(\frac{\sum_{1}^{m}K_{i}H_{i}}{\sum_{1}^{m}H_{i}}-1\right)}$$
where $H_{Q}$ is the roof-cutting height in meters, M is the coal seam thickness, $K_{i}$ is the fragmentation coefficient of the ith layer of rock, and $H_{i}$ is the thickness of the ith layer of rock.

The average thickness of the coal seam in the 20221 working face is 1.88 m, and the actual mining thickness is equal to the height of the roadway, which is 2.8 m. No obvious floor heave was found in similar working faces, and Figure 3 shows that the basic roof lithology is relatively consistent. Therefore, an average fragmentation coefficient of 1.4 was used, and the preliminary design of the top-cutting height was set at 7 m.

#### 2.3.2. Roof-Cutting Angle

Pre-split blasting is performed before mining, and the angle between the pre-split surface and the vertical direction is called the roof-cutting angle (Figure 6). The strength of the immediate roof is relatively low, and it is further weakened due to mining. Therefore, the roof-cutting angle has little influence on the collapse of the immediate roof. However, when the roof-cutting height includes the main roof, if the top-cutting angle is improperly chosen, the rock blocks on the outside of the pre-split surface will still come into contact with and exert pressure on the lateral roof, hindering the smooth collapse of the goaf roof and significantly affecting the stability of the retained entry [26,27,28].

When employing the roof-cutting method in entry-retaining mining, the pre-split surface becomes the critical contact surface for blocks A and B (Figure 6). Only when block B slides and loses stability along the pre-split surface can the smooth collapse of the basic roof strata and the effective cutoff of stress transfer paths be achieved. According to the masonry beam theory and the S-R stability principle of surrounding rock structures [28,29,30], the roof-cutting angle is calculated as follows:(2)$$\theta =\varphi -\tan^{-1}\frac{2(h-\Delta s)}{L}$$
where φ is the friction angle between rock blocks, h is the thickness of the basic roof, L is the length of the basic roof block, and Δs is the amount of subsidence of block B.

The measurement of the mechanical parameters of the basic roof revealed φ to be 27°. Figure 4 shows the location of the weak layer to be at 2–3 m, so Δs is 2 m and h is 3.5 m. The step distance for the collapse of the basic roof in other working faces is approximately L = 10 m. Therefore, the preliminary design of the roof-cutting angle was set at 10°.

### 2.4. Blasting Design

#### 2.4.1. Preliminary Design of Charge Structure

Sandstone has a higher strength, and thus the designed charge volume was 3–4 cartridges. Shale has a lower strength, and the charge volume was 1–2 cartridges. Due to the higher stress at the bottom of the hole, which makes it difficult to split, the bottom charge volume was designed as 3–5 cartridges. The preliminary design of the charge structure was 431 with a total of 8 cartridges, totaling 2.4 kg.

According to the Safety Regulations in Coal Mines [31] and Safety Regulations for Blasting [32], the stemming length should not be less than one-third of the hole depth. The design stemming length was set at 2.5 m.

#### 2.4.2. Interval of Blast Holes

The energy accumulation blasting was simplified, as shown in Figure 7.

To achieve good fracture effects, the blasting-induced cracks of the two holes should be connected. The criterion is that the sum of the damage depths generated by the two energy accumulation blast holes should be greater than the hole interval [1,4,6], which can be expressed as:(3)$$d\geq 2r_{b}\left[1+{\left(\frac{\lambda P_{b}}{(1-D_{0})\sigma_{t}+p}\right)}^{\frac{1}{\alpha }}\right]$$
where $r_{b}$ is the radius of the blasting hole, λ is the lateral pressure coefficient, λ=μ/(1−μ) is the dynamic Poisson’s ratio of the roof rock mass, $D_{0}$ is the initial damage of the rock mass, $\sigma_{t}$ is the tensile strength of the roof rock mass, p is the ground stress of the rock, and $P_{b}$ is the peak pressure of the shock wave on the borehole wall, which can be represented as:(4)$$P_{b}=P_{h}{\left(\frac{r_{e}}{r_{b}}\right)}^{2k}{\left(\frac{L-L_{0}-L_{s}}{L-L_{s}}\right)}^{\gamma }\frac{2}{\left(1+\frac{\rho_{0}}{\rho_{s}}\right)}\xi$$
where $r_{e}$ is the radius of the explosive column, k is the polytropic index ($k=1.9+0.6\rho_{0}$, usually taken as 2), $\rho_{0}$ is the density of the explosive, $\rho_{s}$ is the density of the roof rock mass, L is the depth of the borehole, $L_{0}$ is the air gap length inside the borehole, $L_{s}$ is the blocked length of the borehole, and ξ is the energy accumulation coefficient, which is related to the energy accumulation and is greater than or equal to 1. $P_{h}$ is the initial pressure of detonation, which can be expressed as:(5)$$P_{h}=\rho_{0}uD=\frac{\rho_{0}D^{2}}{k+1}$$
where u is the velocity of motion of the detonation products, and D is the detonation velocity of the explosive.

According to the attenuation law of the detonation stress wave, the calculation equation for the blasting stress damage range *R*_s_ is as follows [4,7]:(6)$$R_{s}=r_{b}{\left[\frac{\lambda P_{b}}{(1-D_{0})\sigma_{t}+p}\right]}^{\frac{1}{\alpha }}$$
where α is the exponent of the attenuation of the detonation stress wave in the rock mass, α=2−μ/(1−μ), which is related to the lithology of the roof and the blasting method. Based on the actual conditions of the Donghuantuo Coal Mine, k = 2, $\rho_{0}$ = 1100 kg/m^3^, and D = 2800 m/s; substituting these into Equation (5) yields $P_{h}$ = 2870 MPa, $r_{e}$ = 35/2 = 17.5 mm, $r_{e}$=55/2=27.5 mm, γ = 2, L = 7.5 m, $L_{0}$ = 2.6 m, $L_{s}$ = 2.5 m, $\rho_{s}$ = 2523 kg/m^3^, and ξ = 10.

Substituting them into Equation (4) yields:

$P_{b}$ = 1510 MPa, μ = 0.23, λ=μ/(1−μ) = 0.3, $D_{0}$=0.6, and $\sigma_{t}$ = 3.156 MPa.

The vertical pressure p=γH = 10 MPa, and α=2−λ = 1.7. According to Equation (3), d = 538 mm. Therefore, the preliminary design of the spacing between blast holes was set at 500 mm.

#### 2.4.3. Initiation Quantity

The number of initiations needs to be adjusted based on the advancing speed of the working face and the concentration of harmful gases. The designed advancement speed of the 20,221 working face was 4.8 m/d, and the spacing between blast holes was 0.5 m, so at least 10 holes need to be blasted each day. Since the concentration of harmful gases is related to factors such as whether the stemming material collapses after blasting, the concentration of harmful gases will be determined through the experiments. To ensure the mining speed, when the concentration of harmful gases exceeds the standard, measures such as increasing the ventilation should be undertaken.

### 2.5. On-Site Blasting Tests

In the Donghuantuo Mine, the roof-caving distance in the same coal seam working face ranges from 42 m to 56 m, and the periodic roof-caving distance ranges from 25 m to 35 m. Considering how the triangular area connecting the open cut and the retained entry influences the collapse of the roof, the test location was set within the range of 60 m to 120 m along the haulage roadway.

Firstly, single-hole blasting tests were conducted to test the charge structure and volume. The single-hole blasting test used the 431 (number represents rolls of explosives) charge structure, as shown in Figure 8. To avoid the influence of adjacent boreholes, the blast holes were arranged at 10 m intervals.

During construction, the positions of each drilling hole were accurately marked along the roof of the roadway on the side of the working face using spray paint based on the centerline of the roadway. Before drilling, it is necessary to inspect the support within a 20 m range of the blasting site, protect the gas and water pipelines, and pre-treat the wire mesh near the cutting seam.

The method of interval charging blasting was adopted, with 3 empty holes separated in the middle. When the hole depth is 7.5 m, a single hole uses 2.5 energy-gathering tubes. The allowed emulsion explosives were installed into the energy-gathering tube in sequence, and then the energy-gathering tube was installed into the blasting hole. Each hole was charged with 2.5 charges in a 431 manner, totaling 8 rolls. The explosive was located at the top of the tube using a fixing block to fix the explosive to the top of the shaped charge tube, and the remaining lower part was empty.

Each shaped charge tube was equipped with a detonator, which was charged in the forward direction. The detonator pin wire and connecting wire of the first shaped charge tube must pass through the second and third shaped charge tubes and be led out. The two energy-gathering tubes were connected with a connecting sleeve. The three energy-gathering tubes were installed into the slit blasting holes in sequence, and all of the slit holes of the energy-gathering tubes must be arranged parallel to the roadway. The bottom of the slit blasting hole was where the initiating explosive roll was located, and the end of the energy-gathering tube hole was filled with gunpowder and fixed with iron wire. The sealing gunpowder was wrapped in damp gunpowder on a kraft paper roll and inserted with a wooden gun rod. The sealing length of the hole should be at least 2.5 m.

Multiple hole-blasting tests were also conducted. Based on the concentration of harmful gases, the pre-split-blasting construction speed, the drilling speed, the charging and blasting speed, production shifts, and the advancing speed of the working face (6 cuts per day, totaling 4.8 m), a design for continuous hole-blasting tests was established. Three sets of tests were conducted, with observation holes set in the middle of each set of borehole distributions. After the continuous hole blasting, the concentration of harmful gases was detected, and borehole observations were made. If the fissure formation rate reached 80% and the concentration of harmful gases did not exceed the requirement, the test was considered successful.

A total of 5 blasting tests were conducted. The fifth test was an analysis and comparison of the results from the previous four tests, which was followed by optimization of the charge parameters. The details of the 5 tests are shown in Table 3.

## 3. Results

### 3.1. Stemming Medium

In the first test, local yellow mud was selected as the stemming material (No. 1) and was prepared on-site. It was found that the preparation speed was slow, and the moisture content of the stemming material was high, making it easily stick to the borehole walls. In the second test, the stemming material (No. 2) was prepared using a stemming machine on the ground, and then transported to the site in plastic wrap. It was found that the stemming material was dry and had obvious particles, and after blasting, the stemming material came out as a whole piece. In the third test, a higher-viscosity red clay (No. 3) was chosen, sieved, and prepared using a stemming machine. After blasting, part of the stemming material came out. In the fourth and fifth tests, a segmented tamping method was used (No. 4), with a tamping length of 250 mm for each individual stemming material segment. After filling, tamping was conducted. With this method, the stemming material no longer came out. It should be noted that the segmented charge structure using a stemming material increased the stemming length and reduced the fissure formation rate at the borehole location.

In the first test, air was used as the interval medium. In the second test, the stemming material was chosen as the interval medium, and it was found to lead to better fissure formation. In the third test, water mixed with the stemming material was used as the interval medium. After blasting, there was less damage to the borehole walls at the charging locations of the pre-split holes, and there were fewer harmful gases and less smoke emitted after stemming material ejection. Although water mixed with the stemming material also increased the stemming length, it was considered to reduce the harmful gases and smoke. Therefore, water mixed with the stemming material was chosen as the interval medium for subsequent tests.

### 3.2. Blasting Parameters

In the fourth test, the same charge volume was used, but different charge structures were compared. It was found that the 3221 charge structure resulted in discontinuous fissure lines, while the 431 charge structure had continuous fissure lines and better fissure formation.

By comparing the third and fourth tests, it was evident that initiating two adjacent pre-split holes resulted in more significant fissure formation compared to initiating two pre-split holes at an interval. This is because during simultaneous hole blasting, the stress waves generated in the adjacent blast holes add up, creating higher compressive stress on both sides of the line connecting the blast holes, which is more conducive to the formation of continuous fissure surfaces.

### 3.3. Fissure Formation

Regardless of the charging method, there were obvious collapse or fragmentation phenomena in the charging zone, as shown in Figure 9. The blasting caused significant damage to the borehole walls in the charging zone.

The uniaxial compressive strength of the 20221 roof rock mass did not exceed 48 MPa, indicating that it is a soft rock mass. Therefore, in the calculation of the fissure formation rate, the length of the stemming segment was not included. The characteristics and fissure formation results of some pre-split holes after blasting are shown in Figure 10.

Based on the borehole observation results, the fissure formation in borehole 149# is shown in Figure 10a. Symmetrical cracks were found in the range from 5 m to 5.2 m after blasting. There were no obvious paired fissures after blasting in borehole 151#, but localized collapse occurred around 2.2 m. Since the stemming material came out completely in both boreholes, most of the blasting energy was released from the borehole and not utilized for tensile fracturing. The fissure formation in borehole #159 is shown in Figure 10b. After blasting, the stemming material came out, and the average fissure formation rate of the borehole was calculated as (300 + 120 + 80 + 3150)/5000 × 100% = 76%. Borehole 161#, which was charged with the 431 structure in segmented stemming, used air as the interval medium. After blasting, the stemming material came out, and the average fissure formation rate of borehole 161# was 94.05%, indicating significant fissure formation. Borehole 163#, which was also charged with the 431 structure in segmented stemming, used the stemming material as the interval medium. A symmetrical fissure was found in the range from 2.8 m to 3.3 m, and the stemming material came out completely.

Borehole 250#, which was charged with the 431 structure in segmented stemming, used water mixed with the stemming material as the interval medium, as shown in Figure 10c. After blasting, the stemming material did not come out. Although cracks were not evident in the range from 2.3 m to 3.3 m, there were continuous yellow fissures along the borehole wall, indicating that the yellow mud was wedged into the fissures. The fissures were primarily distributed symmetrically in the upper-left and lower-right directions in the ranges of 2.3–3.3 m and 4.6–5 m, and paired fissures were mainly observed in the lower-left lower and upper-right directions in the range of 5–6 m. The comprehensive fissure formation rate was calculated as 76%. After blasting, almost no significant fissures were observed in boreholes 250# and 252#, indicating that the presence of the stemming material wedged into the fissures caused this. However, due to poor fissure formation, additional explosives were needed.

Borehole 160# was not blasted, but adjacent boreholes, including 161# with a spacing of 440 mm and 159# with a spacing of 510 mm, were charged and blasted. After blasting, almost no cracks were observed in borehole 160#, but localized wall collapse occurred, as shown in Figure 11, and smoke was present inside borehole 160# after blasting. It was concluded that no continuous fissure was formed within a 440 mm range around the blasting hole, and that only localized weak layers were penetrated.

Figure 12 shows the fissure formation in boreholes 244# and 243#, which were blasted simultaneously. Both boreholes had a 431 structure with segmented stemming, and a wet stemming material was used as the interval medium. The stemming material came out of the borehole, and after blasting, the borehole walls were smooth and bright. In borehole 244#, paired fissures were located in the upper-left and lower-right positions of the borehole wall in the range of 2.5–4.6 m. The average fissure formation rate of borehole 244# was 81.4%, and for borehole 243#, it was 82.4%. A comparative analysis revealed that simultaneous hole blasting resulted in a higher fissure formation rate but also caused more obvious damage to the borehole walls.

Figure 12 also shows the blasting and fissure formation in boreholes 62# and 72#, which were blasted simultaneously. Both boreholes had a 441 structure with segmented stemming, and water mixed with the stemming material was used as the interval medium. The stemming material did not come out, and after blasting, the borehole walls were smooth and bright. Borehole 62# exhibited overall good fissure formation, while borehole 72# had more collapsed sections in the borehole wall. The fissure formation rate was 90% for both boreholes.

Based on the fissure formation in the boreholes, a fissure formation rate distribution map was plotted, as shown in Figure 13. The map indicates a high fissure formation rate ranging from 4.5 m to 7.5 m and exceeding 90%, while the fissure formation rate in the range from 2.5 m to 4.5 m is approximately 50–60%. Due to the influence of the stemming material, the fissure formation rate in the range of 0–2.5 m is almost zero. However, since this part of the rock strata has a lower strength and an obvious weak layer exists in the range from 2 m to 3 m, this section of the roof strata collapses first during mining, and it is not considered when calculating the fissure formation rate.

In the range from 2.5 m to 4.5 m, the fissure formation rate is not high. To increase the fissure formation rate in this range, an additional charge was added to the second energy accumulator, resulting in a 441 charge structure. Based on the analysis of boreholes 62# and 72#, it was found that the fissure formation rate significantly increased in the range from 2.5 m to 4.5 m. This was due to the good fissure formation effect in the range from 4.5 m to 7.5 m, as shown in Figure 14.

Therefore, it was determined that the single-hole charge structure would be 441 with segmented stemming, using water mixed with the stemming material as the interval medium. The charge weight would be 2.7 kg per hole, the stemming length would be 2.5 m, and the spacing between boreholes would be 500 mm.

### 3.4. Post-Mining Monitoring

Figure 15 shows the arrangement of measuring stations for hydraulic supports in the 20221 working face. Fifteen measurement points were set up to monitor the load on the hydraulic supports. Measurement stations were placed at supports 4#, 18#, 32#, 46#, 60#, 74#, 88#, 102#, 116#, and 130#. Additionally, to analyze the impact of the cut-top unloading on the collapse pattern of the roof strata on both sides, extra measurement points were added at supports 8#, 13#, 24#, 121#, and 126#.

Hydraulic shields at different positions were selected from the cutting seam line for mining-induced pressure analysis. The 4# and 13# shields were located at the lower part of the working face, close to the cutting line. The 58# and 67# shields were located in the middle of the working face. The 121# and 130# shields were far from the cutting line. When the hydraulic load is higher than the average mean square deviation of the load, it is determined that there is a roof-weighting phenomenon in the working face [33,34,35,36,37]. The load distribution curve, average load value (dashed line), and pressure benchmark line (solid line) of the shields are shown in Figure 16.

By comparing the roof pressure and the periodic distance in the lower and upper parts of the working face, it can be observed that the upper part of the working face has a greater roof pressure and smaller periodic distance. The main reason for this is that when the monitoring location is far from the fissure, it is less affected by the fissure. In this area, the roof still behaves as a cantilever beam, and the collapse of the goaf cannot achieve a rapid roof-caving state. Therefore, the pressure strength in the working face on the side of the goaf in the lower part where the roof cutting occurs and on the gob side is more significant, and the distance during the periodic roof pressure is relatively small.

Figure 16 shows that there is a clear and sustained roof pressure phenomenon at the position of support 67#. The load of the support in the range of 39.2–56.8 m consistently exceeded the benchmark line for roof pressure determination. In contrast, this phenomenon did not occur at the position of support 13#, indicating that the roof pre-split blasting on the gob side had a significant unloading effect.

The statistics of the roof pressure and periodic distance of the upper and lower supports of the working face during the mining process are shown in Table 4. Compared to the upper part of the working face (support 130#), the lower part (support 4#) had an increased initial abutment pressure distance of approximately 20 m and an increased periodic pressure distance of 3.6 m, while the pressure strength was reduced by about 2 MPa. Comparing the pressure and distance between support 13# and support 4#, it can be found that the closer the working face is to the gob side, the better the unloading effect on the roof, resulting in a lower pressure and larger step distance. Thus, we believe that the collapse of the roof of the working face near the gob side after the roof pre-split blasting is timely.

## 4. Discussion

### 4.1. Comparison with Similar Studies

Both this study and other similar research studies adopted field investigation methods. Initially, preliminary design parameters were determined based on the field conditions and relevant theories. During the engineering process, the roof collapse status was monitored after mining, and the blasting operations were dynamically adjusted based on the analysis of the roof-cutting effect.

From a technical perspective, roof-cutting blasting is meant to effectively disconnect the roof of the roadway from the goaf roof, minimizing the impact of goaf roof collapse on the roadway roof. At the same time, the blasting itself should also minimize damage to the roadway roof. In addition to meeting these requirements, the engineering efficiency should be maximized.

In terms of parameter selection for roof-cutting operations, the existing literature mainly focuses on medium-thick or thick coal seams, with blasting depths ranging from 6 m to 11 m. In this study, the mining height was 3 m, and the preliminary design for the blasting height was 7.8 m. The angle of the cutting line is typically between 0° and 15°, whereas in this study, it was set at 10°. Generally, a smaller fissure angle results in a slower roof collapse, leading to a longer duration and wider range of dynamic load effects caused by the roof collapse. A larger fissure angle would result in a longer hanging roof, increasing the static loading pressure on the sides of the roadway. Therefore, the cutting angle needs to be dynamically adjusted in consideration of the roadway support. The stemming length for blast holes is usually between 1.5 m and 2.3 m, but in this study, the stemming length for cut-top holes was 2.5 m. The main reason for this difference was the high water content observed in most of the roof strata during on-site testing, which necessitated a corresponding increase in stemming length. The charge weight per unit length in other studies ranges from 0.25 kg/m to 0.53 kg/m, while in this study, based on on-site testing, the charge weight per hole was determined to be 0.21 kg. The spacing between blast holes is typically between 400 mm and 600 mm, whereas in this study, it was set at 500 mm.

Compared to the similar literature, this paper provides a detailed description of the preliminary design method for roof-cutting blasting, the steps of our on-site experiments, and the relationship between blasting parameters and post-mining monitoring data. It offers a replicable methodology for field engineering personnel to perform roof-cutting blasting operations.

### 4.2. Limitations and Further Research Directions

This study is a case-based field research study, and the results are dependent on the geological conditions of the site. Additionally, due to budget and time constraints, a further optimization analysis of hole spacing and charge weight was not conducted. Furthermore, limitations in budget and technical expertise prevented the use of a wider range of post-mining monitoring methods, thus hindering the data cross-analysis of the roof collapse status and roadway support strength.

Roof cutting along the goaf has promising prospects due to its environmental friendliness. However, it is still in the stage where engineering experience is being accumulated, and there is a need to establish more practical theoretical analysis methods and detailed engineering technical systems to promote the widespread application of this mining method. From a technical standpoint, the author suggests the development of simpler and more reliable monitoring technologies and equipment, the establishment of a systematic monitoring technical framework, and the automatic analysis of pre- and post-mining data relationships, providing construction personnel with a convenient and reliable decision-making basis.

## 5. Conclusions

Roof blasting is a key operation for entry retaining. This research provides a reproducible construction method for roof-cutting operations and establishes the relationship between blasting parameters and post-mining monitoring data. The following points can be concluded:(1)The preliminary design of the blasting parameters can be achieved based on theories and on-site geological conditions. The roof lithology can be determined using a mine-drilling imaging trajectory detection device. The cutting height and angle can be determined using Equations (1)–(6).(2)Since on-site test methods are provided, the fissure formation mainly depends on the stemming materials, charge structure, and blasting pattern. The effect of different blasting designs can be obtained via fissure formation rate analysis.(3)The effect of pre-split blasting can be determined and adjusted based on post-mining monitoring data. The loading state of the hydraulic shield in the working face can be used to analyze the effectiveness of roof caving in the goaf.

This study contributes to the development of fundamental theories and systematic technical systems for entry retaining via roof cutting, offering high-quality case studies for similar geological engineering projects.

## Figures and Tables

**Figure 1 sensors-23-06391-f001:**
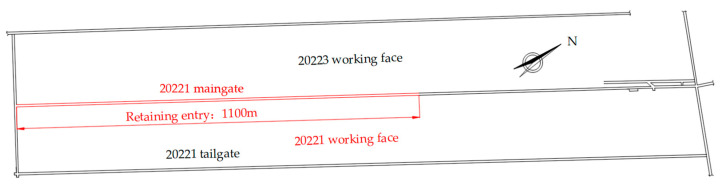
The 20221 working face and retaining entry.

**Figure 2 sensors-23-06391-f002:**
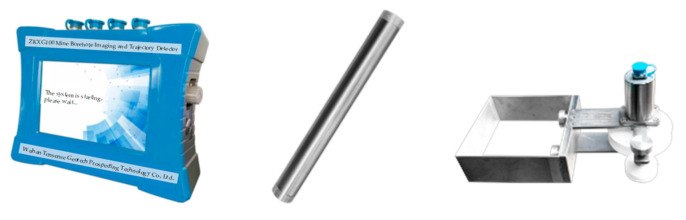
ZKXG100 mine-drilling imaging trajectory detection device; left: host; middle: 24 mm sensor; right: depth detector.

**Figure 3 sensors-23-06391-f003:**
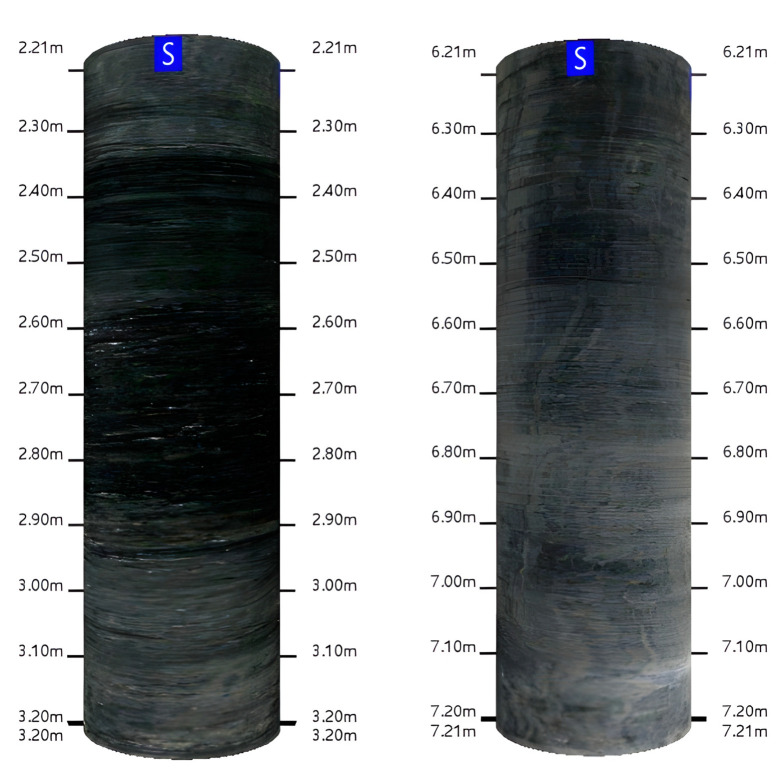
Borehole images at 1500 m.

**Figure 4 sensors-23-06391-f004:**
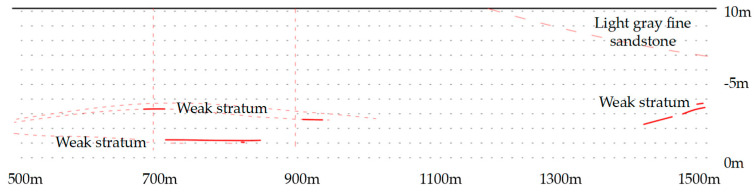
Overlying strata of the 20221 main gate.

**Figure 5 sensors-23-06391-f005:**
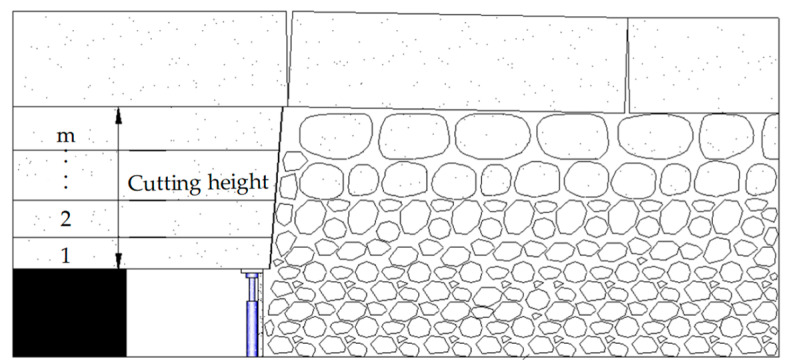
Roof-cutting height design.

**Figure 6 sensors-23-06391-f006:**
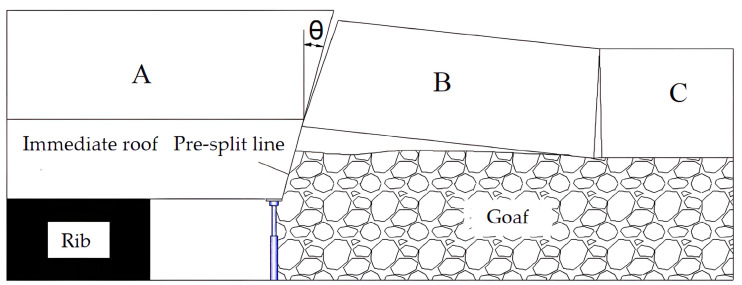
The surrounding rock in the process of entry retaining via roof cutting (A, B, C are the key blocks).

**Figure 7 sensors-23-06391-f007:**
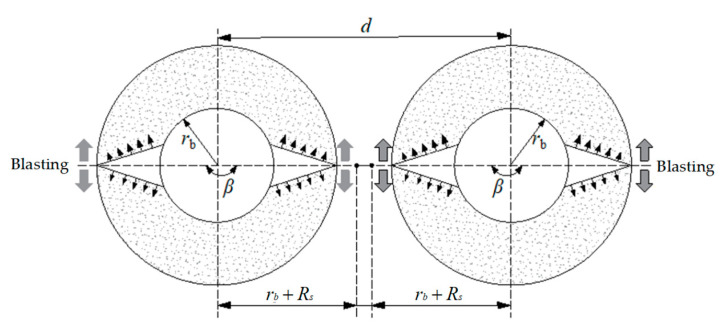
Blasting design model.

**Figure 8 sensors-23-06391-f008:**
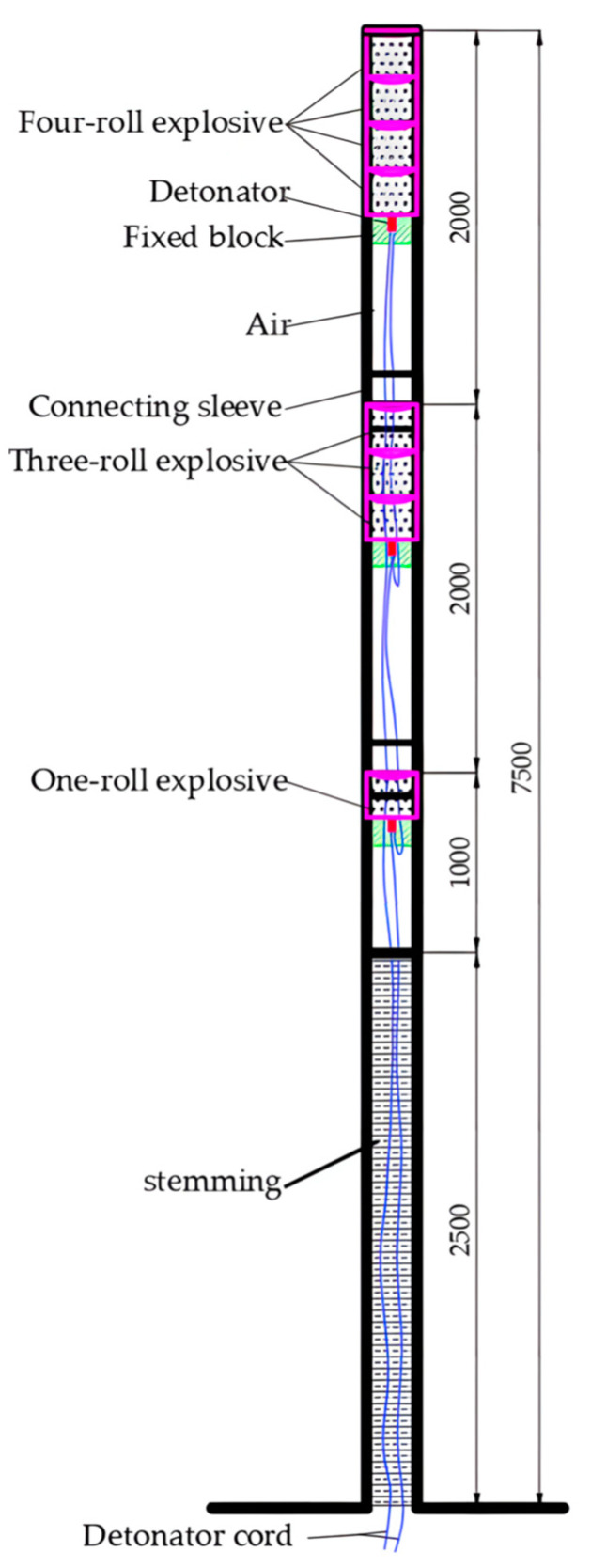
Single-hole blasting charge design (Unit: mm).

**Figure 9 sensors-23-06391-f009:**
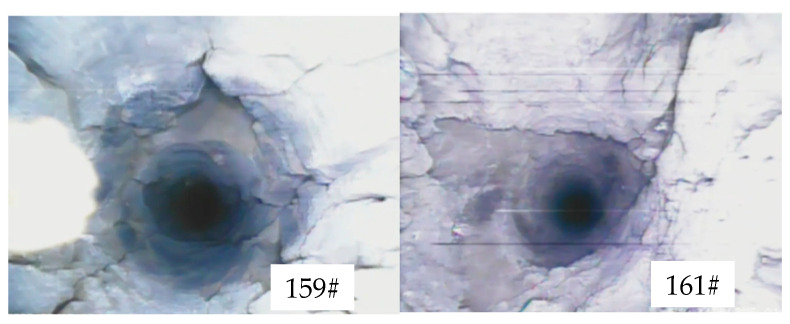
Hole collapse 159# (**top left**), 161# (**top right**), 244# (**bottom left**), and 243# (**bottom right**).

**Figure 10 sensors-23-06391-f010:**
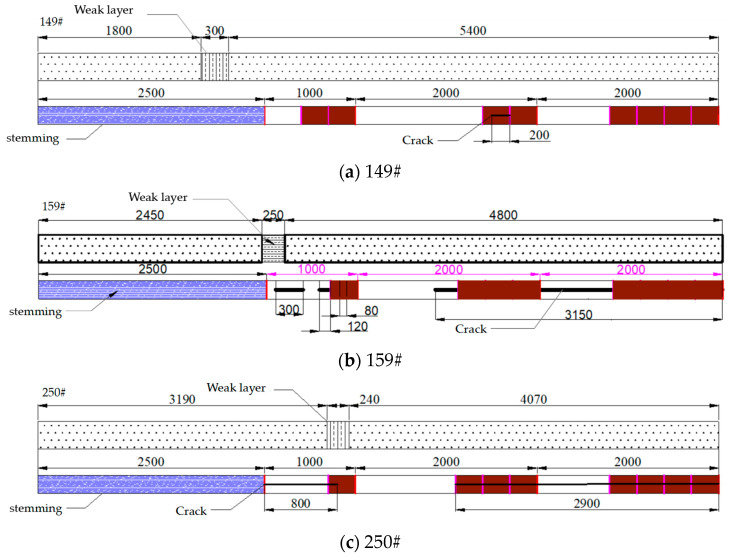
Blasting structure and crack formation of boreholes (**a**) 149#, (**b**) 159#, and (**c**) 250#.

**Figure 11 sensors-23-06391-f011:**
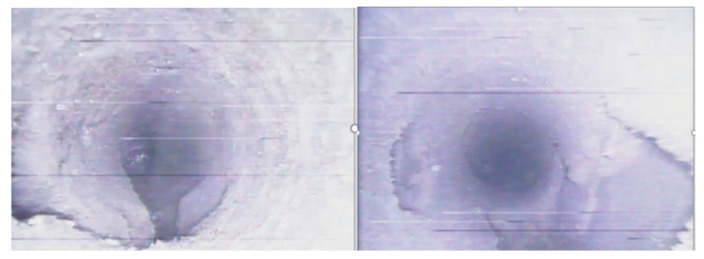
Borehole 160# conditions.

**Figure 12 sensors-23-06391-f012:**
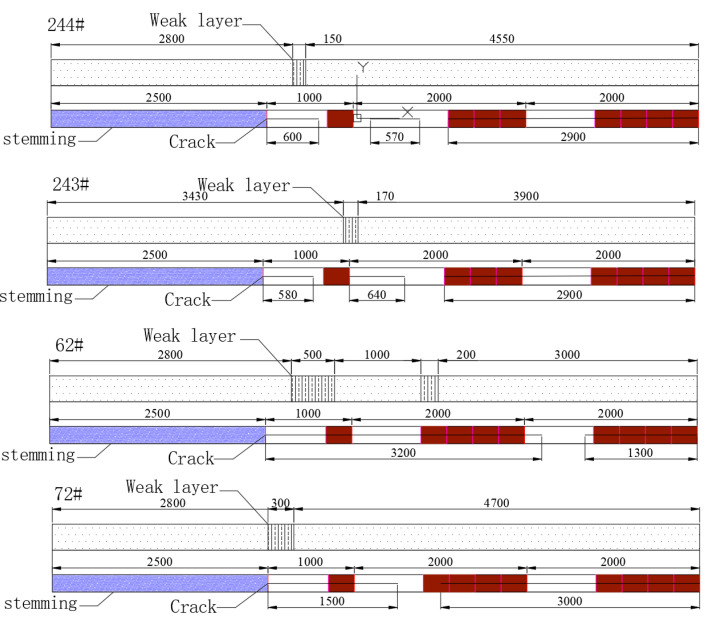
Blasting structure and crack formation of boreholes 244#, 243#, 62#, and 72# (Blue zone represents the stemming section, brown zone represents the charging section).

**Figure 13 sensors-23-06391-f013:**
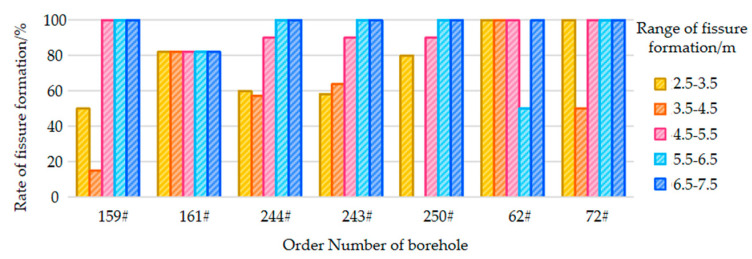
Comparison of fissure formation percentage for different boreholes.

**Figure 14 sensors-23-06391-f014:**
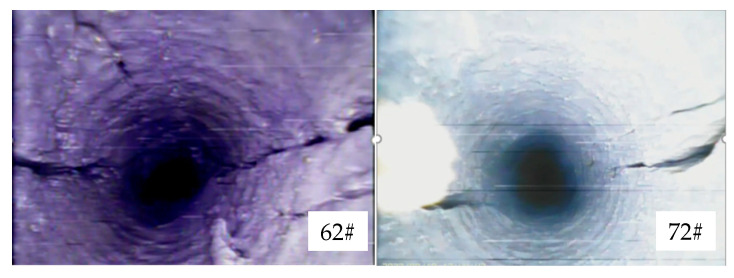
Left: 62# fissure formation at 5.0 m; right: 72# fissure formation at 3.7 m.

**Figure 15 sensors-23-06391-f015:**
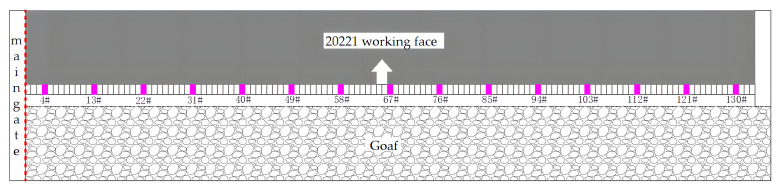
Arrangement of monitoring station.

**Figure 16 sensors-23-06391-f016:**
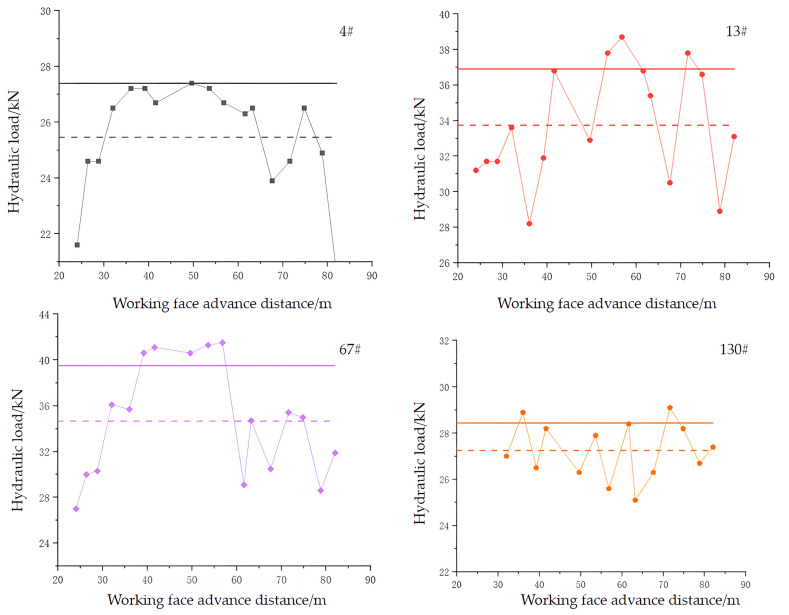
Hydraulic load of shields 4#, 13#, 67#, and 130#.

**Table 1 sensors-23-06391-t001:** Coal seam information.

Thickness (minimum~maximum/average) (m)	1.00~2.76/1.88
Inclined angle (minimum~maximum/average) (°)	16~23/20
Gas (m^3^/min)	0.27~0.34
CO_2_ (m^3^/min)	1.29~2.12
Explosive index (%)	41.50
Spontaneous combustion	Yes

**Table 2 sensors-23-06391-t002:** Surrounding rock information.

Layer	Lithology	Thickness (m)	Features
Main roof	Dark gray siltstone	5.24	Dark gray with uneven fractures, containing a large number of plant fossils such as Koda trees, with siderite nodules and a slightly thin strip-like layer at the bottom.
Immediate roof	Dark gray siltstone	1.75	Dark gray and densely organized, containing siderite nodules and plant fossils, with dark streaks of grayish brown at the top, light streaks at the bottom, and grayish-white streaks containing plant carbonization.
Immediate floor	Dark gray claystone	2.07	Contains plant root fossils, sometimes with thin coal lines, and local siltstone with high carbon content, a 0.25 m thick coal line in the lower part in powder form, with good coal quality.
Floor	Dark gray siltstone	3.39	Stratification is developed, containing plant fossils such as Koda trees, with locally occurring siderite nodules and sometimes thin coal seams.

**Table 3 sensors-23-06391-t003:** Summary of on-site blasting tests.

	Hole	Charge Structure	Stemming (m)	Stemming Material	Observation
1	147	431	2.5 m	1	Stemming collapse
	149	422	2.5 m	1	Stemming collapse
	151	332	2.5 m	1	Stemming collapse
2	159	431	2.5 m	2	Stemming collapse
	160	0			Smoke
	161	431	2.5 m	2	Stemming collapse
	163	431	3 m	2	Stemming collapse
	165	421	2.5 m	2	Stemming collapse
3	243	431	2.5 m	3	Stemming collapse
	244	431	2.5 m	3	Stemming collapse
	250	431	2.5 m	3	
	251	0			Smoke
	252	431	2.5 m	3	
4	241	3221	2.5 m	4	
	240	3221	2.5 m	4	
	236	431	2.5 m	4	
	235	431	2.5 m	4	Collapse
5	72	441	2.5 m	4	
	62	441	2.5 m	4	
	52	441	2.5 m	4	Collapse

**Table 4 sensors-23-06391-t004:** Comparison of abutment pressure and distance.

Shield	Initial Abutment Pressure		Periodic Abutment Pressure	
	Magnitude/MPa	Distance/m	Magnitude/MPa	Distance/m
130#	29.5	29.0	28.5	17.6
13#	38.7	41.0	37.8	14.8
4#	27.4	49.4	26.5	21.2

## Data Availability

The data used to support the findings of this study are available from the corresponding author upon reasonable request (chenying@lntu.edu.cn).

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
