# Peer review of "Preliminary Design and On-Site Testing Methodology of Roof-Cutting for Entry Retaining in Underground Coal Mine"

_sensors, 2023, doi:10.3390/s23146391_

Round 1
Reviewer 1 Report
In the reviewer‘s opinion, this paper is very interesting and can contribute to community. The paper can be accepted after addressing some minor comments:
1. Figure 1 is not clear. The blue、red and black lines should be explained. Also, Figure 4 is not clear.
2. Equation1, what are the differences between m and M?
3. More quantitative results should be presented.
Author Response
Attachment

Reviewer 2 Report
This paper contains interesting field research that relates to new mining method. The performed field detection as well as calculation of roof blasting parameters are interesting due to the obtained preliminary designing, in particular for various blasting configurations. I recommend this article for acceptance for publication in Sensors after implementing the necessary corrections.
1. The English language should be polished thoroughly. It is better to use a professional language editing institute, in such case a language editing certificate should be attached along with authors’ response letter.
2. In abstract, introduction and conclusion, the significant of the research should be stated clearly.
3. In the abstract, some information about field blasting test method and result should be added.
4. Line 56: top cutting?
5. Fig. 1: Make sure all lines, values and wording are readable. Please use standard size and font type for all wordings and values at all figures. Suggest to use Palatino Linotype (type font), text size 9. Please check the whole manuscript and do correction accordingly.
6. Table 1: capital/lowercase should follow the requirements of the journal.
7. It is suggested to eliminate Fig. 2.
8. Fig. 4: Make sure all lines, values and wording are readable. Please use standard size and font type for all wordings and values at all figures. Suggest using Palatino Linotype (type font), text size 9. Please check the whole manuscript and do correction accordingly.
9. Some statements in the paper need to be unified, such as: “formula, equations, Eq”, “working face/panel”, etc.
10. Fig. 7: Make sure all lines, values and wording are readable. Please use standard size and font type for all wordings and values at all figures. Suggest using Palatino Linotype (type font), text size 9. Please check the whole manuscript and do correction accordingly.
11. Eq 3: please follow the requirement of the journal. Please check the whole manuscript and do correction accordingly.
12. Lines 217-221: please re-write.
13. Fig. 8 should be redrawn or eliminated.
14. Table 3: capital/lowercase should follow the requirements of the journal.
15. Table 3: collapse means eruption?
16. All subtitles, please follow the requirement of the journal.
17. Fig. 10: Make sure all lines, values and wording are readable. Please use standard size and font type for all wordings and values at all figures. Suggest using Palatino Linotype (type font), text size 9. Please check the whole manuscript and do correction accordingly.
18. In the subsection 4.2, please verify lines 439-454.
19.The cited references are too old, and it is recommended to add the last three years of literature.
Author Response
Attachment

Reviewer 3 Report
This paper is well structured, has an average significant and novelty. The topic covered is absolutely interesting. The aim of the work is very interesting, the introduction is complete and updated on the topic. The results are very clear. The conclusions are consistent with the evidence and arguments presented addressing the main question that are posed.
Author Response
Attachment
